# Performance and Microbial Community of Different Biofilm Membrane Bioreactors Treating Antibiotic-Containing Synthetic Mariculture Wastewater

**DOI:** 10.3390/membranes10100282

**Published:** 2020-10-14

**Authors:** Huining Zhang, Xin Yuan, Hanqing Wang, Shuoqi Ma, Bixiao Ji

**Affiliations:** 1College of Civil Engineering and Architecture, Zhejiang University, Hangzhou 310000, China; 21812200@zju.edu.cn; 2NingboTech University, Ningbo 315000, China; wanghanqing@smedi.com (H.W.); m1904096014@163.com (S.M.); jibixiao@nit.net.cn (B.J.); 3Ningbo Research Institute, Zhejiang University, Ningbo 315100, China

**Keywords:** tetracycline, norfloxacin, mariculture wastewater, fixed biofilm AO-MBR, microbial community, membrane fouling

## Abstract

The performance of pollutant removals, tetracycline (TC) and norfloxacin (NOR) removals, membrane fouling mitigation and the microbial community of three Anoxic/Oxic membrane bioreactors (AO-MBRs), including a moving bed biofilm MBR (MBRa), a fixed biofilm MBR (MBRb) and an AO-MBR (MBRc) for control, were compared in treating antibiotic-containing synthetic mariculture wastewater. The results showed that MBRb had the best effect on antibiotic removal and membrane fouling mitigation compared to the other two bioreactors. The maximum removal rate of TC reached 91.65% and the maximum removal rate of NOR reached 45.46% in MBRb. The addition of antibiotics had little effect on the removal of chemical oxygen demand (COD) and ammonia nitrogen (NH_4_^+^-N)—both maintained more than 90% removal rate during the entire operation. High-throughput sequencing demonstrated that TC and NOR resulted in a significant decrease in the microbial diversity and the microbial richness MBRs. *Flavobacteriia*, *Firmicutes* and *Azoarcus*, regarded as drug-resistant bacteria, might play a crucial part in the removal of antibiotics. In addition, the dynamics of microbial community had a great change, which included the accumulation of resistant microorganisms and the gradual reduction or disappearance of other microorganisms under antibiotic pressure. The research provides an insight into the antibiotic-containing mariculture wastewater treatment and has certain reference value.

## 1. Introduction

In recent years, mariculture has become a fast-developing industry with increasing global population and seafood demand [1]. China is the world’s largest mariculture country and is accompanied by the abuse of a large number of antibiotics [2]. In mariculture wastewater, numerous antibiotics are overused to prevent bacterial contamination, cure diseases and promote fish growth [3,4,5]. Nevertheless, only 20–30% of antibiotics used in mariculture systems are absorbed by mariculture products, while residual antibiotics (>75%) are left over in the feeding water and accumulate in the sediment [6]. The improper treatment of mariculture wastewater may cause the deterioration of the surrounding estuary ecosystem [7]. Studies have shown that the antibiotics in mariculture wastewater could affect the ecotoxicity of aquatic organisms in the surrounding estuarine ecosystems [8,9]. Antibiotics will inhibit the activity of microorganisms in the biological sewage treatment system, thereby affecting the removal of organic and nitrogen-containing compounds. Antibiotics can also severely affect biological systems and livestock manure treatment systems [10,11,12]. The widespread use of antibiotics in mariculture environments has caused new contamination of antibiotic resistance genes (ARG). Therefore, antibiotics as emerging contaminants in mariculture wastewater are of increasing concern to both researchers and the general public.

Due to the high cost of physical–chemical treatment, biotechnology is an alternative method to treat maricultural wastewater. Biosorption and biodegradation are the key processes of antibiotic removal in maricultural wastewater and degradation of organic pollutants. In bioreactors, antibiotics would be adsorbed, biodegraded and hydrolyzed [13]. It has been proposed that the adsorption of activated sludge was the major removal mechanism for fluoroquinolones, such as norfloxacin [14,15] and tetracycline [13,16]. The chemical properties of antibiotics and the wastewater treatment process are used to determine the effectiveness of antibiotic removal. The membrane bioreactor has already been utilized as a useful option to remove antibiotics from wastewater compared to the conventional water treatment processes [17,18]. Biofilm can improve pollutant removal through high biomass density and rich biophase [19]. The presence of TC and NOR in mariculture wastewater might affect the removal of COD and nitrogen in BF-MBR. The BF-MBR performance was closely related to the diversity and richness of the microbial in the biofilm. Consequently, it was significant to study whether antibiotics in the wastewater have an adverse impact on the microbial diversity and richness of the microbial communities in biological systems. There were a few studies that reported that the activity and composition of drug-resistant bacteria might play a crucial part in the removal of antibiotics.

Because of its broad-spectrum activity against both Gram-positive and Gram-negative bacteria, TC and NOR were widely used in mariculture systems, so they were chosen as the typical representatives for antibiotics study. Moreover, a novel BF-MBR with the fiber bundle bio-carriers (MBRb) was designed to improve performance in a high-salt mariculture environment. Two MBRs were selected for comparison in this study: MBRa with suspended bio-carriers to represent a system with moving biofilm, and MBRc without bio-carriers. We previously explored the removal effects of conventional contaminants in these MBRs with different salinities [20]. There were few studies that reported the performance of these MBRs treating antibiotic-containing synthetic mariculture wastewater in a high-salt mariculture environment.

The primary aim of this research was to provide theoretical basis and technical support for the choice of MBR treatment process of mariculture wastewater containing antibiotics. The indicators of bioreactor pollutant removal, system stability, membrane fouling and microbial community structure were examined in synthetic wastewater distribution to simulate mariculture wastewater of 30 g/L under 200 μg/L antibiotic stress.

## 2. Materials and Methods

### 2.1. Reactors and Operating Conditions 

The three reactors (designed as MBRa, MBRb, and MBRc) were setup in parallel in this study (Figure 1). In all bioreactors, the experiments were carried out in aerobic tanks (12 L) and an anoxic tank (6 L). The ultrafiltration membrane module was from Lanjinquan, Shanghai Yule Industry and Trade Co., Ltd., China MBRb was equipped with two fiber bundle carriers (Yulong, Jiangsu, China) consisted of a rope wrapped with a polyvinylidene fluoride fiber bundle. The suspended carriers (Yulong, Jiangsu, China) in MBRa were hollow cylinders made of polypropylene. The effective specific surface area of these carriers is 1200 m^2^/m^3^. The details and packing density of the fiber bundle bio-carrier and suspended bio-carrier, and the operating conditions were described in a previous study [20]. The pre-anoxic tank in three AO-MBRs could improve denitrification ability. During the operation, a large amount of biomass was adsorbed onto the biofilm of the fiber bundle carrier. Compared to the MBRc, the reactors with carriers had higher biomass content, which means that bio-carriers improved the growth of microorganisms. The sludge reflux ratio from aerobic tank to anoxic tank was 150%. Three MBRs were operated under a sludge retention time (SRT) of 52 days, with a hydraulic retention time (HRT) of 18 h. Dissolved oxygen in the aerobic tank was 2.7 (±0.5) mg/L.

### 2.2. Inoculated Sludge and Wastewater Composition

The test sludge was inoculated from a sewage disposal plant in Ningbo, China. In the experiment, the synthetic mariculture wastewater was introduced by an influent pump into the MBRs. The composition of the synthetic mariculture wastewater was listed in the document [20] and 30 g/L of salinity was selected, the usual salinity level in mariculture wastewater. The influent concentrations of COD, NH_4_^+^-N and NO_3_^−^-N were 100–140, 8–10 and 3.3–4.2 mg/L, respectively. The prepared TC and NOR standard stock solutions with concentrations of 100 mg/L were stored in a refrigerator at 4 °C, protected from light. The appropriate amount of standard storage solution (its initial concentration was 100 mg/L and diluted to 200 μg/L) were added to the influent, which was slightly higher than the concentration of antibiotics contained in the Xiangshan mariculture wastewater.

To study the effect of antibiotics on the MBRs process in mariculture wastewater, the experiment was divided into 3 stages: no antibiotics were added in the influent in phase 0 (P0) (0–11 day), then TC at the concentration of 200 μg/L was spiked into the influent in phase 1 (P1) (12–27 day), finally in order to explore the coexistence of different antibiotics the NOR was added to the influent with TC with the concentration of 200 μg/L in phase 2 (P2) (28–52 day), respectively.

### 2.3. Analytical Methods

The samples were selected daily and each sample was analyzed in three replicates. All the water samples were analyzed immediately (within 2 h) for parameters, including those with COD, NH_4_^+^-N, NO_3_^−^-N and NO_2_^−^-N with the Standard Methods [21] using UV-Vis Spectrophotometer (D500, Hach, Loveland, CO, USA). 

Solid phase extraction (SPE) (SPEQ-24B, Shanghai, China) was used as the pretreatment method of samples and the methodology of sample pretreatment was adopted from earlier studies [22,23]. The TC and NOR were quantified using high-performance liquid chromatography (HPCL, Hitachi CM5000, Tokyo, Japan) with ultraviolet detection at 270 nm and 280 nm, respectively, which was equipped with an Agilent Eclipse XDB-C18 chromatography column (4.6 × 150 mm^2^, 5 μm). The mobile phase for TC consisted of 81%:19% (*v*/*v*) acetonitrile and 0.1% formic acid at a flow rate of 1 mL/min. The detailed information of analysis conditions of HPLC for NOR was described before [24]. 

In addition, a comprehensive analysis of microbial communities under different stages by 16S rRNA sequencing were performed. First, activated sludge in the aerobic tanks of MBRa, MBRb, and MBRc were sampled when their removal performances were stable at each stage. The E.Z.N.A.^®^ Soil DNA Kits (Omega Bio-tek, Norcross, GA, USA) were used to extract DNA from the sludge samples. High-throughput sequencing was selected by an Illumina MiSeq platform in accordance with the report [20].

## 3. Results

### 3.1. Reactor Performance

#### 3.1.1. Removal of Bulk Pollutants

As shown in Figure 2a, the addition of TC and NOR had little effect on the COD removal compared to the reactor performance in phase 0 (without antibiotics). The average COD removal rate was maintained at 91.24 ± 1.35%, 95.13 ± 0.66% and 90.21 ± 1.66% in phase 0, and 88.89 ± 1.32%, 95.57 ± 1.82% and 88.01 ± 2.18% in phase 1, and 90.29 ± 1.40%, 95.05 ± 1.47% and 88.57 ± 1.95% in phase 2 in MBRa, MBRb, and MBRc during the entire operation, respectively. The results indicate that all reactors had the high removal efficiency of organic matter, and the COD removal efficiency of the MBRb reactor was a bit higher than that of the MBRa reactor and MBRc reactor throughout the operation. The results were consistent with those of previous studies, indicating that ≤20 mg/L antibiotics had no effect on the elimination of COD [25].

The antibiotics effects on the NH_4_^+^-N removal rate in the MBRa, MBRb, and MBRc system are shown in Figure 2b. Before antibiotics were added (0−11 day), the NH_4_^+^-N removal rate of MBRa and MBRc was similar (95.55 ± 3.03% MBRa and 94.91 ± 2.96% MBRc, *P* > 0.05), while NH_4_^+^-N removal rate MBRb (97.03 ± 2.05% MBRb, *P* > 0.05) was slightly higher than those of MBRa and MBRc. After adding 200 μg/L TC in the influent, the removal rate of NH_4_^+^-N in effluent fluctuated obviously in a short period of time, even fell below 0%, then recovered quickly and declined to 91.5 ± 1.45% in MBRa, 93.33 ± 1.32% in MBRb and 90.17 ± 2.28% in MBRc, respectively. The literature shows that high concentrations of tetracycline did not have a significant negative effect on dehydrogenase activity and the tetracycline inhibits the microbial biomass of NOB [26]. The internal organism nitrogen is released into the solution, increasing the concentration of ammonia nitrogen [27]. When NOR was added at 200 μg/L in the influent, which already contained TC, the removal rate of NH_4_^+^-N was greatly affected. This may be that the subsequent denitrifying flora was inhibited from the beginning and gradually restored its activity as the running time increased. The final removal rate of NH_4_^+^-N was maintained at 88.7 ± 2.79% MBRa, 91.47 ± 2.62% MBRb and 88.46 ± 2.35% MBRc, respectively. Additionally, the effluent concentrations of NO_2_^−^-N (Figure 2c) rose gradually in the reactors, which might be related to the presence of TC and NOR that led to nitrite accumulation [28]. MBRb and MBRa had better NO_3_^−^-N removal rate (above 50%) (Figure 2d) in the presence of antibiotics than MBRc because the thicker biofilm further enhanced denitrification capacity.

These results indicated that carbon and nitrogen shared similar trends at every stable stage, which means that the 200 μg/L concentration of TC and NOR tested did not affect organic matter removal and nitrogen removal efficiency. However, the concentration of TC and NOR reported to affect the nitrification performance was much higher than that reported in mariculture wastewater (ng/L to μg/L) and used in this study [29,30].

#### 3.1.2. Removal of Antibiotics

After a brief period of acclimatization, the water quality indexes of the effluent tended to relatively stable values. When antibiotics were added to the three reactors, some quantities of antibiotics were decomposed, some quantities of antibiotics remained in the wastewater, and some quantities of antibiotics entered the activated sludge due to adsorption.

Figure 3 showed the removal of antibiotics in the three reactors at different operating stages. The three reactors shared a similar trend. The removal rates of TC first increased from 73.86% to 89.77% in MBRa, 78.15% to 91.65% in MBRb and 62.36% to 81.07% in MBRc at 16 days, and then quickly decreased to 78.0% in MBRa, 83.01% in MBRb and 63.25% in MBRc at 22 days. Note that the removal rates of TC rose sharply in the first 10 days. The reason might be that adsorption played a main role in the early process. The adsorption of TC occurred quickly at the start and almost achieved the maximum adsorption capacity after 10 days [31]. With the increase in time, microorganisms gradually adapted to TC in the system, and then its biodegradation dominated later. Subsequently, the removal rate of TC decreased with the addition of NOR. The performance of MBRa and MBRb was superior to MBRc, and MBRb showed the best effect on TC removal. The importance of biodegradation on the removal of this antibiotic was also shown [32], since they suggested that the removal of TC was characterized by a rapid adsorption and slow biodegradation process. These systems were operated by having very long SRT and acclimation potential, which might contribute to the degradation process of TC. The removal rate of TC was very high during the whole experiment.

The effluent concentration of NOR changed greatly during the whole operation period. At the initial stage of adding NOR, the removal rate of NOR was higher due to the adsorption of NOR by sludge. However, with the prolongation of time, the active sites of sludge were gradually occupied, activated sludge adsorption gradually reached an equilibrium, the removal rate of NOR began to decrease, and the final removal rates remained at 40.36 ± 0.2% of MBRa, 46.46 ± 0.11% of MBRb and 28.65 ± 0.18% of MBRc, *P* < 0.05, which indicated that the reactors still had a certain removal ability of NOR. Kummerer et al. [33] examined the effects of broad-spectrum antibiotics on microorganisms in activated sludge. He found that only a small fraction of bacteria in sludge were affected. This was because the main removal of the quinolone antibiotics in MBRs was through the adsorption and biodegradation of activated sludge, and mainly based on the adsorption of sludge [34]. The removal rate of NOR was much lower than TC, which was likely because the fact that the adapted sludge had been saturated with TC and thus decreased their adsorption potentials.

### 3.2. Membrane Fouling Behavior during Continuous Operation

As an important indicator of membrane fouling propensity, the development of the transmembrane pressure (TMP) was monitored during the operating period. The time course changes of TMP in each MBRs were illustrated in Figure 4. Physical cleaning of the membranes (manually wiping with a sponge) was performed when TMP exceeded 35 kPa.

As shown in Figure 4, the MBRa had nine-times TMP up to a total of 35 kPa, which was far more than that of MBRb (two times) and MBRc (four times) throughout the operation. It was revealed that the MBRb using fiber bundle bio-carriers could reduce membrane fouling and the membrane fouling rate of MBRa was even more serious than that of MBRc. This finding was consistent with Yang et al. [35], who reported that the bioreactor with suspended carriers (MB-MBR) showed a higher cake layer resistance than conventional MBR (CMBR) due to the presence of a large amount of filamentous bacteria inhabited in suspended solids in MB-MBR, resulting in the membrane fouling rate of MB-MBR about three times that in CMBR. The TMPs of MBRb increased slowly in the first 10 d, followed by a plateau where the TMP stabilized around 8.06 kPa for over 9 d and eventually rose to 35 kPa on day 26. The membrane filtration cycle was about eight times longer than that of MBRa. The operation of another membrane filtration cycle was performed on the MBRb and similar TMP development was observed. This might be due to the fact that fiber bundle biologically thicker biofilm carrier provided a better ability to resist antibiotic impact, creating a more abundant biodiversity and more stable environment, so as to improve the degradation of the membrane fouling agent. Therefore, it was presumed that the MBRb had a better control in alleviating membrane fouling.

### 3.3. Microbial Community Dynamics

The activated sludge of properties and microbial community determined the stability of system and performance of treatment. Thus, the microbial community structures of the sludge in the reactor were analyzed.

#### 3.3.1. Microbial Diversity and Richness in the Three MBRs 

The microbial sludge communities’ diversity and richness of MBRs during the operation phase were analyzed by high-throughput sequencing. Good’s coverage of each sample was over 0.99, meaning that the obtained sequence libraries covered the microbial diversity of MBRs. As shown in Table 1, the Shannon and Chao1 index values in MBRb were higher than those in the MBRa and MBRc during the entire operation. The Chao1 indexes could reflect the richness of the microbes, and the Shannon indexes could reveal the diversity of microbes [36]. It demonstrated that the thicker fixed biofilm in MBRb was more conducive to develop a stable living environment for various microorganisms.

The Shannon diversity index values gradually decreased from 4.14, 4.91, and 3.84 for MBRa, MBRb, and MBRc to 2.75, 3.70, and 2.57 for MBRa, MBRb, and MBRc, respectively, suggesting that TC and NOR could reduce the microbial diversity of MBRs. The Chao 1 richness index values decreased from 2971.25, 3526.4, and 2741.59 for MBRa, MBRb, and MBRc to 824.08, 992.04, and 800.43 for MBRa, MBRb, and MBRc, respectively. The variations in the Chao 1 indexes demonstrated that the existence of TC and NOR could influence the microbial richness of MBRs. These results were in agreement with previous studies [37,38]. Although the first addition of TC caused a sharp decline in microbial diversity, the richness of the species varied slightly with the extension of exposure time and the addition of NOR, which indicated that the microbial community was no longer as susceptible as the initial one. More simply, the community became more adaptive to antibiotics, which was similar to previous studies [39,40]. 

#### 3.3.2. Microbial Community Structure Analysis

The microbial community structure of the samples was analyzed in three stages. Sample C represented the activated sludge without adding antibiotics in MBRc, and samples MBRa-1, MBRb-1, MBRc-1, MBRa-2, MBRb-2, and MBRc-2 represented the activated sludge with (1) TC and (2) TC and NOR of the three reactors.

At the phylum level, it revealed that the two predominate phyla were *Proteobacteria* (70.99%, 52.42%, 49.01%, 37.75%, 57.3%, 55.36% and 51.5%) and *Bacteroidetes* (13.32%, 38.88%, 30.84%, 37.36%, 30.55%, 22.39% and 33.99%) in MBRa-1, MBRb-1, MBRc-1, MBRa-2, MBRb-2, and MBRc-2, respectively (Figure 5a). Despite the fluctuation, total relative abundance of *Proteobacteria* and *Bacteroidetes* in the communities increased dramatically after TC and NOR addition, indicating that members from these predominant lineages largely made up the resistant species proliferated in the MBRs. However, the number of *Proteobacteria* decreased in phase 2 was smaller to that in the phase 1, meaning that it had resistance to the antibiotics with the increase in operation time. On the contrary, the *Acidobacteria* was inhibited by the antibiotics and, in the activated sludge after the addition of the TC and NOR, the *Acidobacteria* gradually disappeared, manifesting that the *Acidobacteria* could not be adapted to an antibiotic-containing environment. The other two microbial phyla, *Actinobacteria* and *Planctomycetes*, seemed to be sensitive to the presence of antibiotics, when NOR was added, the relative abundance of *Actinobacteria* and *Planctomycetes* decreased from 2.4%, 13.37% and 3.87%, 17.53% in MBRb-1 and MBRc-1 to 2.16%, 11.33% and 1.05%, 9.91% in MBRb-2 and MBRc-2, respectively, except for MBRa. It was worth noting that *Firmicutes* in samples MBRb-1 and MBRb-2 was much higher than other MBRs. This demonstrated the *Firmicutes* was capable of adapting the antibiotics and was favored in the activation of sludge for resisting antibiotics. As previously reported on the microbial communities, *Firmicutes* had a key part in the process of complex substances [41,42].

The changes in structure and relative abundance of microorganisms on the class level are shown in Figure 5b. The dominant class varied among samples, the two predominating classes were *Flavobacteriia* (22.01%, 22.16%) and *Betaproteobacteria* (21.5%, 32.36%) in MBRa-1 and MBRa-2, respectively, *Flavobacteriia* (19.01%, 28.92%) and *Gammaproteobacteria* (19.28%, 20.69%) in MBRb-1 and MBRc-2, respectively, *Flavobacteriia* (27.41%) and *Alphaproteobacteria* (19.73%) in MBRc-1 and *Alphaproteobacteria* (22.22%) and *Gammaproteobacteria* (15.66%) in MBRb-2. Previous results have shown that at higher concentrations of antibiotics, these microbes had a greater proportion of relative abundance, indicating that they had drug-resistance [43,44]. In past studies, *Alphaproteobacteria* were able to carry on the sulfur metabolism, remove organic matter and nitrogen, which were the main functions of biofilters in maricultural recirculation systems [45] and *Betaproteobacteria* played a significant role in the removal of nitrogen [46,47]. The abundance of *Flavobacteriia* and *Gammaproteobacteria* in reactors fed with wastewater containing antibiotics was significantly higher than those in the P0. Pollet et al. [48] found that *Flavobacteriia* might be the key actor in the functioning of marine biofilms. Moreover, Li et al. [49] found that the tetracycline-resistant bacteria isolated from the treatment of oxytetracycline production wastewater by a sewage treatment plant were mainly *Gammaproteobacteria*, followed by *Alphaproteobacteria* and *Betaproteobacteria*.

To analyze bacteria community clearly on a genus level, some major genera were displayed in the relative abundance heatmap using the species abundance matrix to draw. At the P1 and P2 stage, the predominant genera were *Arenibacter* (18.44%, 19.2%, 13.22% and 28.02%), *Azoarcus* (20.48%, 31.69%, 12.3% and 17.39%) and *Vibrio* (8.96%, 2.61%, 10.13% and 19.27%) in MBRa-1, MBRa-2, MBRb-2 and MBRc-2, respectively, while in MBRb-1, *Arenibacter* (17.12%), *Phycisphaera* (11.97%), *Vibrio* (11.68%) and *Azoarcus* (9.08%) and in MBRc-1, *Arenibacter* (24.32%), *Phycisphaera* (17.03%), *Azoarcus* (7.64%)and *Labrenzia* (5.41%) were most abundant. The relative abundance of *Azoarcus* increased from 20.48% at P1 to 31.69% at P2 in MBRa, 9.08% at P1 to 12.30% at P2 in MBRb and 7.64% at P1 to 17.39% at P2 in MBRc, respectively. It was reported that *Azoarcus* had the function to degrade aromatic compounds [50]. Thus, *Azoarcus* in this study might play an important part in the degradation of NOR, since NOR was a kind of aromatic compound containing three benzene rings. Moreover, *Martelella* abundance was 4.7% and 7.28% in MBRb-1 and MBRb-2, which was almost non-existent in both MBRa and MBRc. *Martelella* had the ability to degrade certain polycyclic aromatic hydrocarbon (PAH) compounds, such as acenaphthene, fluorene, phenanthrene, anthracene, and used the poisons as the sole carbon source [51].

## 4. Conclusions

The results revealed that, in the presence of antibiotics, all three MBRs were able to maintain a good removal in COD and NH_4_^+^-N. TC and NOR were removed through different removal mechanisms; however, MBRb was more advantageous compared to the other two reactors under the same antibiotics level and had better performance on membrane fouling mitigation. In addition, the existence of antibiotics resulted in a significant decrease in the microbial diversity and the microbial richness, which showed some obvious changes at different stages in different MBRs. *Flavobacteriia*, *Firmicutes* and *Azoarcus*, regarded as drug-resistant bacteria, might play a crucial part in the removal of antibiotics. In addition, three different levels (phylum, class and genus) of the dynamics of microbial community had a great change, which included the accumulation of resistant microorganisms and the gradual reduction in or disappearance of other microorganisms under antibiotic pressure.

## Figures and Tables

**Figure 1 membranes-10-00282-f001:**
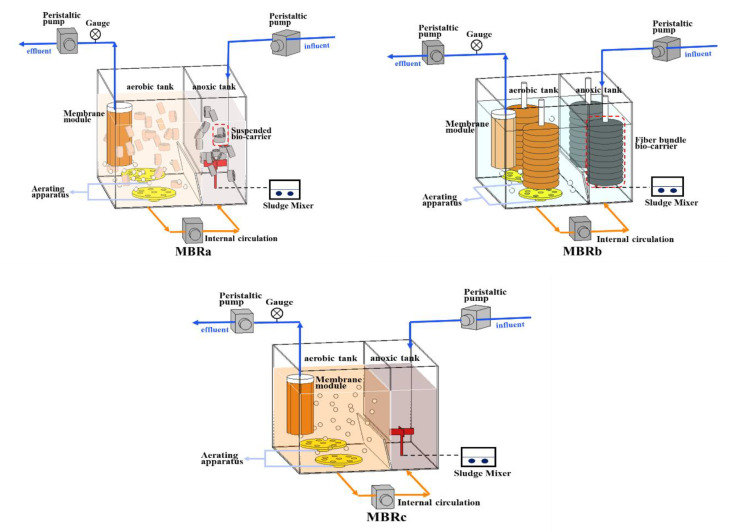
Configurations of membrane bioreactors MBRa, MBRb, and MBRc.

**Figure 2 membranes-10-00282-f002:**
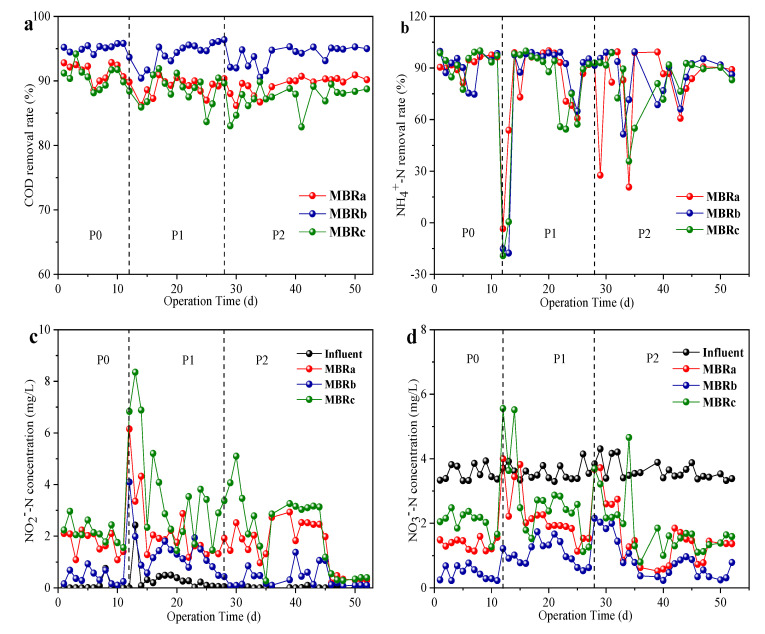
Removal performance of different reactors in different operating phases, COD removal rate in (**a**), NH_4_^+^-N removal rate in (**b**), NO_2_^−^-N influent and effluent components in (**c**) and NO_3_^−^-N influent and effluent components in (**d**).

**Figure 3 membranes-10-00282-f003:**
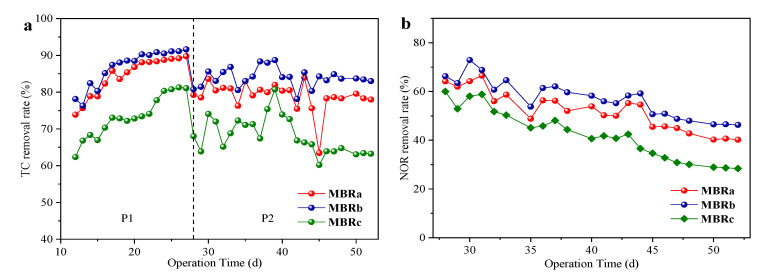
TC removal rates (**a**) and NOR removal rates, (**b**) in reactors at stable state in P1 and P2.

**Figure 4 membranes-10-00282-f004:**
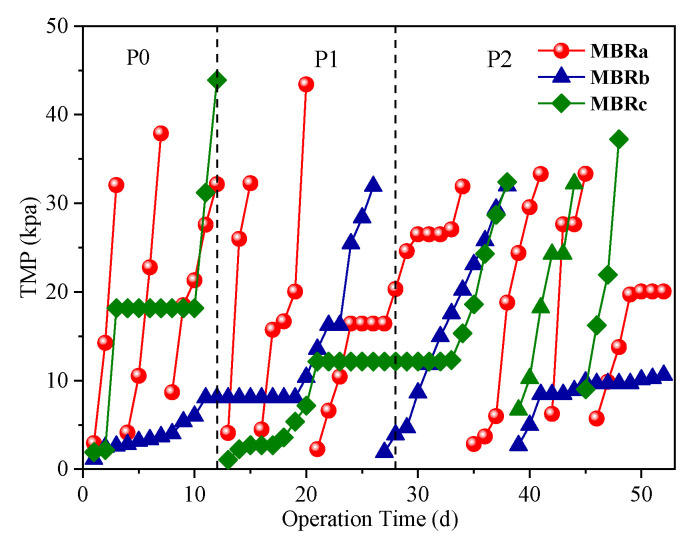
Variations in transmembrane pressure (TMP) in each MBRs.

**Figure 5 membranes-10-00282-f005:**
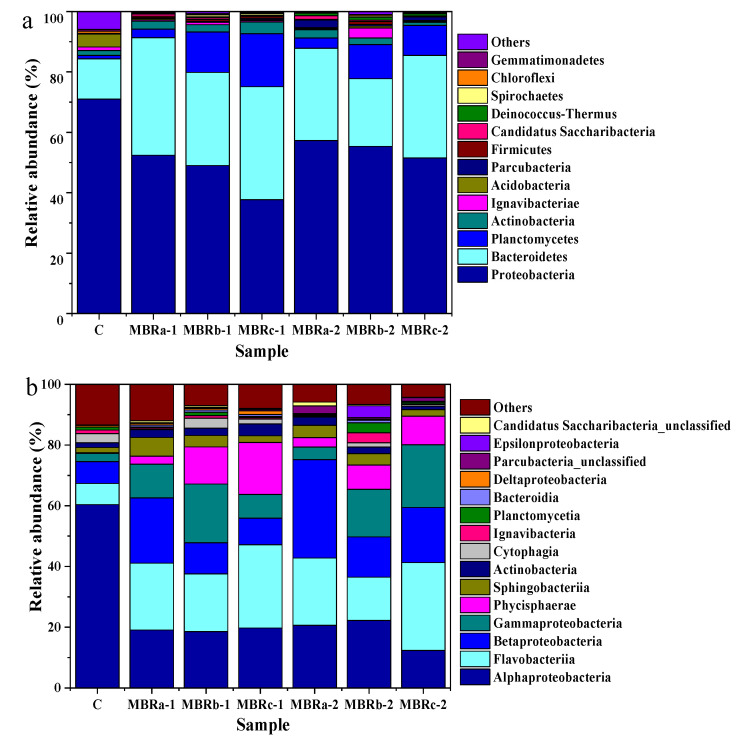
Microbial community structure in sludge samples in P0 (C) in MBRc, and P1 (MBRa-1, MBRb-1 and MBRc-1) and P2 (MBRa-2, MBRb-2 and MBRc-2). Results are shown at the (**a**) phylum level and (**b**) class levels.

**Table 1 membranes-10-00282-t001:** The Shannon and Chao1 index of activated sludge samples in three stages in MBRa, MBRb, and MBRc.

	Shannon Index			Chao1 Index			Good’s Coverage
	MBRa	MBRb	MBRc	MBRa	MBRb	MBRc	All Reactors
P0	3.84	4.91	4.14	2741.59	3526.4	2971.25	≥0.98
P1	3.00	3.62	3.16	1279.82	1590.43	1398.06	1.00
P2	2.75	3.70	2.57	824.08	992.04	800.43	≥0.99

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
