# Peer review of "Performance and Microbial Community of Different Biofilm Membrane Bioreactors Treating Antibiotic-Containing Synthetic Mariculture Wastewater"

_membranes, 2020, doi:10.3390/membranes10100282_

Round 1
Reviewer 1 Report
The document describes the effect of the presence of antibiotics in the degradation of artificial marine wastewater in a membrane bioreactor. The question being asked is important since one of the priorities nowadays is the elimination of emerging pollutants, including antibiotic residues.
Three different membrane bioreactors are tested, with differences on the modules used. After a period of acclimatization, the wastewater is spiked with one antibiotic, and later, a second antibiotic is added.
The experimental set up is appropriate, and the variables under study are also useful in answering the research question. The document is descriptive since no statistical analysis is done to assess the three bioreactors' performance differences. Please consider using some statistical analysis to backup your conclusions.
On the other hand, there is no information on the number of replicates done for each experiment or measure. Since there are drastic variations in the removal of nitrogen, for example (Figure 2b, 2c), it is important to show that this is reproducible. The indication of variation (std deviation) will increase the graphics' information, so please consider a cleaner form of presenting the data.
In figure 4, to clarify, was the MBRa backwashed to eliminate the biofouling. I do not remember seeing the information on the document.
Figure 5 has a lot of information. Figure 5c is difficult to read, and the genera's names cannot be properly identified. Also, as with the previous results, it is important to clarify if the description and the conclusions are based on one analysis.
Author Response
Thank you very much for your helpful suggestions for our manuscript entitled, “Performance and microbial community of different biofilm membrane bioreactors treating antibiotic-containing synthetic mariculture wastewater”. Your suggestions enabled us to both improve our research and revise our manuscript. We have resubmitted a revised manuscript with revisions marked.
Our responses to the reviewers’ comments are below.
Major Flaws:
Point 1: The experimental set up is appropriate, and the variables under study are also useful in answering the research question. The document is descriptive since no statistical analysis is done to assess the three bioreactors' performance differences. Please consider using some statistical analysis to backup your conclusions. On the other hand, there is no information on the number of replicates done for each experiment or measure. Since there are drastic variations in the removal of nitrogen, for example (Figure 2b, 2c), it is important to show that this is reproducible. The indication of variation (std deviation) will increase the graphics' information, so please consider a cleaner form of presenting the data.
Response 1: Thank you so much for catching this. The samples were selected daily and each sample was analyzed in three replicates (Line 111 in the MS). Because the amount of data is too large, using std deviation will make the chart more crowded. In addition, the results of each sample analysis in three replicates did not differ much.
Point 2: In figure 4, to clarify, was the MBRa backwashed to eliminate the biofouling. I do not remember seeing the information on the document.
Response 2: Thank you. The MBRa was not backwashed to eliminate the biofouling. Physical cleaning of the membranes (manually wiping with a sponge) without backwashing to eliminate the biofouling was performed when TMP exceeded 35 kPa. The time course changes of TMP in each MBRs were illustrated in Figure 4.
Point 3: Figure 5 has a lot of information. Figure 5c is difficult to read, and the genera's names cannot be properly identified. Also, as with the previous results, it is important to clarify if the description and the conclusions are based on one analysis.
Response 3: So sorry we were vague here. According to your opinion, we also think that the existence of this figure does not mean much to the main idea of the whole manuscript. So, we have deleted Figure 5c in the revised manuscript.
Reviewer 2 Report
This study investigates the performance of pollutant removals, tetracycline and norfloxacin removals, membrane fouling mitigation, and microbial community of three MBRs. The authors found that MBRb had the best effect on antibiotic removal and membrane fouling mitigation compared to the other two bioreactors. The work is largely sound and covers the detailed study of three MBRs. The manuscript is well written and logically laid-out. Thus, in my opinion, this manuscript may be accepted for publication with Membranes, after minor revision.
Comments:
- The introduction can be strengthened by clarifying the novelty of the study.
- Is there any specific reason to select tetracycline and norfloxacin antibiotics for this study? Kindly clarify and include it in the manuscript. (The reason may be tetracycline and norfloxacin are broad-spectrum antibiotics that are active against both Gram-positive and Gram-negative bacteria)
- The discussion is too broad. Revise it to support the results obtained.
- The conclusion section must contain a summary of the study and not the background of the study (delete line 320-322).
Author Response
Thank you very much for your helpful suggestions for our manuscript entitled, “Performance and microbial community of different biofilm membrane bioreactors treating antibiotic-containing synthetic mariculture wastewater”. Your suggestions enabled us to both improve our research and revise our manuscript. We have resubmitted a revised manuscript with revisions marked.
Our responses to the reviewers’ comments are below.
Major Flaws:
Point 1: The introduction can be strengthened by clarifying the novelty of the study.
Response 1: Thank you for your suggestion. In this study, a novel BF-MBR with the fiber bundle bio-carriers (MBRb) was designed to improve performance in a high-salt mariculture environment. Two MBRs were selected for comparison in this study: MBRa with suspended bio-carriers to represent a system with moving biofilm, and MBRc without bio-carriers. There were few studies that reported the performance of these MBRs treating antibiotic-containing synthetic mariculture wastewater in a high-salt mariculture environment. The introduction was strengthened by clarifying the novelty of the study in the revised manuscript.
Point 2: Is there any specific reason to select tetracycline and norfloxacin antibiotics for this study? Kindly clarify and include it in the manuscript. (The reason may be tetracycline and norfloxacin are broad-spectrum antibiotics that are active against both Gram-positive and Gram-negative bacteria).
Response 2: Thank you, once again. We have clarified the reason in the revised manuscript (L65). Because of its broad-spectrum activity against both Gram-positive and Gram-negative bacteria, TC and NOR were widely used in mariculture systems, so they were chosen as the typical representative for the antibiotics study.
Point 3: The discussion is too broad. Revise it to support the results obtained.
Response 3: Thank you, once again. We streamlined the discussion, and specifically deleted statements that are not too relevant to the topic of the paper in the revised manuscript. For example, we deleted Figure 5c and the discussion.
Point 4: The conclusion section must contain a summary of the study and not the background of the study (delete line 320-322).
Response 4: Thank you for your suggestion. We have deleted Line 320-322.
Reviewer 3 Report
I suggest that the authors revise the manuscript. It has many scientific and non-scientific errors. For example, the authors did not discriminate between adsorption and biodegradation. So, some conclusions raised me doubts. I highlighted the ones I detected and attached my comments / suggestions. From summary to conclusion, hard work is needed to improve the quality of work.

Round 2
Reviewer 1 Report
My questions have been incorporated, I would just recommend the authors, in further research, to not consider "slight" variation of results, as a basis for not using statistical analysis. Since the results presented are the result of one sample, analyze in triplicate as for certainty of the analytical result, this should not be considered as variation in the system, but a variation on the analytical measurement. Therefore, std as requested in Figure 2, should not be included.
Reviewer 3 Report
The manuscript has been improved after referees comments. However, the subject is not new. Even though, it is well organized and the ideas are appropriate.